# ASSESSING THE IMPACT OF DISTRIBUTION SHIFT ON REINFORCEMENT LEARNING PERFORMANCE

## ABSTRACT

Research in machine learning is making progress in fixing its own reproducibility crisis. Reinforcement learning (RL), in particular, faces its own set of unique challenges. Comparison of point estimates, and plots that show successful convergence to the optimal policy during training, may obfuscate overfitting or dependence on the experimental setup. Although researchers in RL have proposed reliability metrics that account for uncertainty to better understand each algorithm's strengths and weaknesses, the recommendations of past work do not assume the presence of out-of-distribution observations. We propose a set of evaluation methods that measure the robustness of RL algorithms under distribution shifts. The tools presented here argue for the need to account for performance over time while the agent is acting in its environment. In particular, we recommend time series analysis as a method of observational RL evaluation. We also show that the unique properties of RL and simulated dynamic environments allow us to make stronger assumptions to justify the measurement of causal impact in our evaluations. We then apply these tools to single-agent and multi-agent environments to show the impact of introducing distribution shifts during test time. We present this methodology as a first step toward rigorous RL evaluation in the presence of distribution shifts.

## 1 INTRODUCTION

The field of RL has enjoyed some spectacular recent advancements, like reaching superhuman levels at board games (Silver et al., 2016; 2018; Bakhtin et al., 2022), sailboat racing (McKinsey & Company, 2021), multiplayer poker (Brown & Sandholm, 2019), and real-time strategy games (Berner et al., 2019; Vinyals et al., 2019). Transitioning RL toward a rigorous science, however, has been a more complicated journey. Like other fields in machine learning, progress in RL might be compromised by a lack of focus on reproducibility combined with more emphasis on best-case performance. To remedy this problem, reliability metrics have been proposed to improve reproducibility by making RL evaluation more rigorous. Some examples include the dispersion and risk of the performance distribution (Chan et al., 2020), and interquartile mean with score distributions (Agarwal et al., 2021). Past work, however, does not assume the presence of distribution shift during test time.

In general, distribution shift in machine learning occurs when there is a difference between the training and test distributions, which can significantly impact performance when the machine learning system is deployed in the real world (Koh et al., 2021). In supervised learning, distribution shift could cause a decrease in accuracy. We will focus on distribution shift in RL, which could cause a decline in expected returns. This impact on performance is a symptom of overfitting in deep RL, which is a problem that requires carefully designed evaluation protocols for detection (Zhang et al., 2018). While there are many types of distribution shift, we will focus on test-time adversarial examples and the introduction of new agents in multi-agent ad hoc teamwork.

Taking inspiration from car crash tests and safety ratings from trusted organizations, like the ratings standards of the National Highway Traffic Safety Administration (NHTSA) or the Insurance Institute for Highway Safety (IIHS), we believe RL would benefit from techniques that evaluate robust performance after training. Here, we contribute some recommendations for evaluation protocols using time series analysis to measure the performance of RL agents that encounter distribution shift at test time. Specifically, we recommend (1) the comparison of time series forecasting models of agent performance, (2) using prediction intervals to capture the distribution and uncertainty of

future performance, and (3) counterfactual analysis when distribution shift has been applied by the experimenter. We believe this stress-test methodology is a promising start towards reliable comparison of pretrained RL agents exposed to distribution shift in both single and multi-agent environments.

In section 2, we mention past related work on distribution shift and RL reproducibility. In section 3, we argue why time series analysis is needed for evaluation of RL algorithms under distribution shift. In section 4, we outline our recommendations for RL evaluation with time series analysis. In section 5, we provide examples of such analyses in single and multi-agent RL. In section 6, we conclude with suggestions for future research in RL evaluation and describe how it can advance ML safety and regulation.

## 2 RELATED WORK

Distribution (or dataset) shift occurs when the data distribution at training time differs from the data distribution at test time (Quinonero-Candela et al., 2008). The focus of this paper is not to detect distribution shift (Rabanser et al., 2019), but to assume it exists while measuring agent performance. There has been work on reproducibility under distribution shift for supervised learning (Koh et al., 2021), but RL will be the focus here. In particular, we focus on how overfitting to the training environment can affect the agent's performance during evaluation in a test environment (Zhang et al., 2018). Although we are taking a time series perspective in this paper, we are not proposing a new method of machine learning to train time series models (Ahmed et al., 2010; Masini et al., 2023). Also, we are not proposing a method of causal machine learning (Peters et al., 2017). That is, we are not advocating a model that learns causal representations. Furthermore, we do not claim to be the first to use interventions in simulations for RL, or other areas of machine learning (Ahmed et al., 2021; Lee et al., 2021; Verma & Srivastava, 2021; Lee et al., 2023; Verma et al., 2023). We contribute a methodology that includes counterfactual time series analysis, using the impact visualization strategy from Brodersen et al. (2015), as a way to measure the impact of distribution shift after training. We will use time series and counterfactual analysis only as methods of RL performance evaluation. For clarity, what we propose is not a technique to train time series or causal machine learning models, but a methodology for RL evaluation and reliability on models that have already been trained.

Any distribution shift can be used in our methodology. Due to the interest and research progress in adversarial attacks and ad hoc teamwork, the distribution shifts we investigate are adversarial attacks on images (Atari game observations) and agent switching in multi-agent environments. There has been extensive research on adversarial attacks on supervised learning models, like support vector machines and neural networks (Huang et al., 2011; Biggio et al., 2012; Goodfellow et al., 2014; Kurakin et al., 2016). While we will focus on the adversarial attacks proposed in Huang et al. (2017), there has been related research on adversarial attacks in single-agent (Kos & Song, 2017; Pattanaik et al., 2017; Rakhsha et al., 2020; Zhang et al., 2020), and multi-agent RL (Gleave et al., 2019; Ma et al., 2019; Figura et al., 2021; Fujimoto et al., 2021; Casper et al., 2022; Cui et al., 2022). Another set of experiments will test the ad hoc teamwork of the group of agents (Stone et al., 2010; Barrett & Stone, 2015; Rahman et al., 2021; Mirsky et al., 2022), where agent switching will be treated as a distribution shift among the group.

Rigorous evaluation of RL algorithms is still a topic that requires further investigation. Henderson et al. (2018) show that even subtle differences, like random seeds and code implementation, can affect the training performance of deep RL agents. Engstrom et al. (2020) give a more thorough study of RL implementation and provide evidence that seemingly irrelevant code-level optimizations might be the main reason why Proximal Policy Optimization (Schulman et al., 2017) tends to perform better than Trust Region Policy Optimization (Schulman et al., 2015). Colas et al. (2018) show how the number of random seeds relates to the probability of statistical errors when measuring performance in the context of deep RL. The inherent brittleness in current deep RL algorithms calls into question the reproducibility of some published results. This has led to proposals for rigorous and reliable RL evaluation techniques grounded in statistical practice. Chan et al. (2020) recommended reliability metrics like interquartile range (IQR) and conditional value at risk (CVaR). Jordan et al. (2020) suggested metrics like performance percentiles, proposed a game-theoretic approach to quantifying performance uncertainty, and developed a technique to quantify the uncertainty throughout the entire evaluation procedure. Agarwal et al. (2021) recommended stratified bootstrap confidence intervals, score distributions, and interquartile means. For multi-agent cooperative RL, Gorsane et al. (2022)

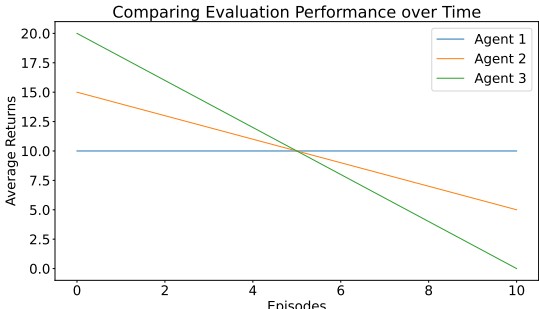

Figure 1: In this simplified plot of agent performance in the presence of worsening distribution shifts over time. All three agents have average returns of 10. It is clear, however, that agent 3 is the least desired agent over time. Even though agent 3 starts out with the highest average returns, it seems to have overfit to the training environment and fails to maintain its superior performance. Point estimates alone would not capture this behavior.

proposed a standard performance evaluation protocol using RL recommendations from past papers. What sets our methodology apart from past work is the emphasis on time series analysis of RL performance with distribution shift during test time. Chan et al. (2020) has included after learning metrics in their work, but focuses on performance variability from trained policy rollouts alone, which is unlikely to capture performance in the presence of significant environment changes.

## 3 THE NEED TO MEASURE PERFORMANCE OVER TIME

### 3.1 A TIME SERIES PERSPECTIVE

We argue for time series analysis as a solution to the problems that point estimates and confidence intervals alone cannot fix. Because we are not certain of the agent's future performance, forecasting the agent's performance is needed if we assume RL agents will inevitably encounter distribution shifts in its environment after training. Point estimates alone can fail to explain decreases in performance at test time. In Figure 1, we assume three agents that act in an environment that experiences increasing distribution shift as the number of episodes continues. Each episode can be interpreted as a day's performance of an RL agent. This is motivated by a hypothetical scenario where RL developers are deploying their agents in an environment that experiences changes they did not anticipate. All agents achieve the same average returns, but there is a clear difference of performance over time. In this hypothetical example, Agent 3 seems to have overfit to the training environment because it starts high but ends as the lowest performer. In the longer term, Agent 2 is preferable because its decrease in performance is slower, and eventually outperforms Agent 3. Agent 1 represents the ideal RL agent because its performance over time never decreases.

As shown in previous work (Chan et al., 2020; Agarwal et al., 2021), confidence intervals are important to quantify the uncertainty of point estimates of aggregate performance. In evaluating performance in the presence of distribution shift, however, confidence intervals for mean scores are insufficient. If we want to account for the uncertainty of the performance trends, it would be preferable to understand the distribution of values and where we expect our forecasting model to generate the next data point sampled (Buteikis, 2020). Hence, when evaluating trends in performance under distribution shift, prediction intervals can be more helpful than confidence intervals. Time series analysis can also be used to measure the causal impact of distribution shift on one (or many) agents. One way to accomplish this is counterfactual analysis. That is, we need a model of the agent acting in the environment as if the change had never taken place. In our methodology, we will assume the trained agents have achieved a flat (slope = 0) trend in performance (as opposed to noisy or random) in the absence of distribution shift at test time.

Within-episode performance is difficult to measure because it requires knowledge of the particular environment's states to know when to make an intervention. To make our methodology more general, we measure the impact of distribution shift over many episodes, where $X^A(t)$ is the expected

performance at an episode $t$. This allows us to measure the positive (or negative) impact of distribution shifts over time. An example of this could be measuring the energy usage or customer satisfaction of a RL-trained HVAC system each day over a week if exposed to distribution shift.

## 3.2 RL AND THE FUNDAMENTAL PROBLEM OF CAUSAL INFERENCE

Practitioners of causal inference must remember its fundamental problem: If we were to expose a unit $u$ to some intervention, we will never see the world where we never exposed the unit $u$ to that intervention (Holland, 1986). That is, observing individual treatment effect is impossible. This has not discouraged researchers from developing methods that circumvent this problem (Spirtes et al., 2000; Pearl, 2009; Imbens & Rubin, 2015; Peters et al., 2017). These methods usually require assumptions that manage the messiness of the real world, which makes it possible to construct valid arguments in favor of causal inference. RL agents in simulated environments, however, allow us to make assumptions that might be too strong for less predictable units (like humans or animals). In a deterministic environment, with a given random seed, regardless of how many times you reach a state $s$, the agent will choose the same action and receive the same reward at state $s$. That is, RL agents in deterministic environments with fixed random seeds allow us to circumvent the fundamental problem of causal inference because we can just reset the environment and see what happens when we choose to intervene or not. This does not contradict Henderson et al. (2018), where they argue that different random seeds can lead to different performances and behaviors. Here, we claim that different RL agents with their own random seed may exhibit different behaviors, but will repeat those behaviors if the random seeds are fixed. Now that we are concerned with RL evaluation over time, we can make the following assumption:

**3.1 RL Fixed Seed Assumption.** *Let $T$, such that $1 \leq T \leq N$, be the time an intervention occurs. Let $G$ designate groups such that $G = 1$ indicates the treatment group and $G = 0$ indicates the control group. Consider a time series outcome $X^A(t)$, where $A$ is a RL agent in a simulated deterministic environment $\mathcal{M}$. Let $X^A(t < T)$ be the performance before time T. Then,*

$$\mathbb{E}[X^A(t < T)|G = 1] = \mathbb{E}[X^A(t < T)|G = 0] \tag{1}$$

*Let $X^A_{U=u}(t)$ be the performance measurement as above with the counterfactual intervention $U = u$. We define $U = 1$ and $U = 0$ as being exposed to an intervention (e.g., distribution shift) and not exposed, respectively. Then, on average, the performance of the control group is the counterfactual performance of the treatment group as if it had not been exposed to the intervention:*

$$\mathbb{E}[X^A(t)|G = 0] = \mathbb{E}[X^A_{U=0}(t)|G = 1] \tag{2}$$

This assumption basically says that if we have a fixed set of random seeds[1] in a deterministic environment, then, on average, the expected returns of the control group is the same as the expected returns of the treatment group if no out-of-distribution intervention ever occurred. This makes intuitive sense because both groups have the same random seeds, which implies they will exhibit identical behavior in deterministic environments if no outside influence is introduced. This is helpful when justifying causal inference in the Appendix.

## 4 RECOMMENDATIONS FOR TIME SERIES EVALUATION

### 4.1 IF YOU CAN CONTROL WHEN THE DISTRIBUTION SHIFT OCCURS, MEASURE THE CAUSAL IMPACT

Similar to car crash tests in controlled settings, we propose an evaluation method where the experimenter can control when a RL agent experiences a shift in distribution. First, take a trained RL agent and have it interact in its environment until some time (or episode) $T$, which will be the pre-treatment period. At time $T$, the post-treatment period starts with the experimenter applies a shift in distribution. Such distribution shifts include adversarial examples (Goodfellow et al., 2014). In the multi-agent cooperative tasks, sudden replacement of agents can also cause shifts in distribution (Mirsky et al., 2022). The experimenter logs the performance scores at each point in time. When the agent reaches

---

[1]Reproducibility can be difficult to achieve when running on a GPU. We discuss this further and how to control for it in the Appendix.

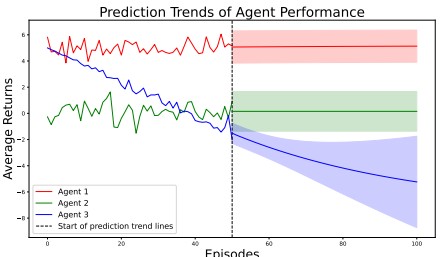 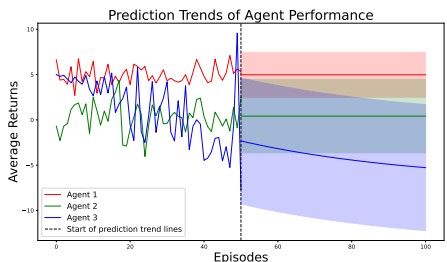

Figure 2: The idea behind these graphs is the same as in Figure 1. The main differences are that we use time series forecasts and prediction intervals to show the predicted performance trend of each agent. Here, the performances between episodes 0-50 are the measured performances of the agents in the environment. The plots at episodes 50-100 are not measured performance, but prediction trends of future performance with prediction intervals. **Left:** The differences in performance are clear because the prediction intervals do not overlap at the end of the plot. Hence, Agent 1 has the best performance because the forecast is not decreasing and the prediction interval is small. The fact that Agent 2's interval does not overlap with Agent 3's prediction interval shows that Agent 1 and Agent 2 have significantly better performance forecasts. **Right:** Here, all agents have noisier performance. Even though Agent 3 still has a downward trend, its much noisier performance briefly spikes up to match Agent 1's performance. Hence, we want prediction intervals that anticipate this uncertainty by showing interval overlap between agent performance over time. There is no longer a significant difference between Agents 2 and 3 because their prediction intervals overlap at every time step.

the end of the experiment (or some time threshold), evaluate the causal impact from the counterfactual model (the RL agent control group) and the treatment group's post-treatment performance using difference-in-differences (DiD) (Cunningham, 2021; Huntington-Klein, 2021). The results will show how much the distribution shift impacted the performance of the agent assuming a counterfactual model that represents the agent's performance if the distribution shift never happened.

To show the impact of the distribution shift, we use the template of time-series impact plots from Brodersen et al. (2015). This template consists of three panels: an original plot, a pointwise plot, and a cumulative plot. In the original plot, the raw performance is shown. The pointwise plot shows the difference between the observed data and the counterfactual predictions, which is the inferred causal impact of intervention. The cumulative plot shows the cumulative impact of the intervention over time. Unlike in Brodersen et al. (2015), instead of Bayesian structural time-series models, we use DiD because we are measuring only one variable (returns) and our assumptions justify its application. The justification for DiD in this methodology is provided in the methods section of the Appendix.

### 4.2 OTHERWISE, COMPARE AGENTS USING SIMPLE TIME SERIES TRENDS WITH PREDICTION INTERVALS

Here, we assume the experimenter has no control over when the distribution shift will occur. It is also possible that multiple instances of distribution shift can occur at different times. Such scenarios are intended to be a closer representation of real-world RL agent deployment. This implies, without further assumptions, an observational study is the best we can do. We propose comparison of RL performance using time series trends, like Holt's Linear Damped Trend Method (Gardner Jr & McKenzie, 1985; Holt, 2004; Hyndman & Athanasopoulos, 2018), with prediction intervals. An example is provided in Figure 2. Here, we can compare the performance of agents trained on different RL algorithms. The idea is similar to the comparison of agents in Figure 1. The main difference is that we are not just focusing on the agents' measured performance, but also on the forecast of future performance with prediction intervals. Here, robust performance can be interpreted as the the forecast of future performance and the prediction intervals visualize its uncertainty over time. Ideally, the agent that performs the best would have the highest trend line and the smallest prediction interval. Using time series forecasts with prediction intervals are meant to be an improvement over point estimates with confidence intervals to better show the uncertainty of measured RL performance over time. Since we will be using Holt's linear damped trend method, 95% prediction intervals might be too narrow (Hyndman, 2014). Hence, we will default to 99% prediction intervals. Like in section 3,

we interpret each time point as an episode, and $X^A(t)$ represents the RL agent's performance during that episode. Further description of this method is provided in the methods section of the Appendix.

In a style similar to Gorsane et al. (2022), we summarize the previous recommendations into the following protocol:

---

**A Protocol for Measuring Performance in the Presence of Distribution Shift**

**Input:** Environments $E_1 \ldots E_K$, Algorithms $A_1 \ldots A_M$ with corresponding time series metric $X^A(t)$, Distribution Shifts $d_1 \ldots d_N$.

**1. Evaluation parameters - defaults**

- Number of episodes for impact and observational plots.
- Number of episodes for forecasting with prediction intervals ($= 100$).
- Number of random seeds/runs ($= 10$ from Agarwal et al. (2021)).

**2a. If you can control when the distribution shift occurs, measure the causal impact**

- Measure the causal impact of each distribution shift intervention $d_n$ using difference-in-differences with plots of the (i) raw performance, (ii) pointwise impact, and (iii) cumulative impact (from Brodersen et al. (2015)).
- For each pair $(E_k, d_n)$, algorithm $A_i$ performs better than $A_j$ in the presence of $d_n$ if $A_i$'s cumulative impact of average returns is higher than $A_j$'s at the end of the experiment.

**2b. Otherwise, compare agents using time series trends with prediction intervals**

- In the presence of a distribution shift, plot a forecast, using Holt's Linear Damped Trend Method, with 99% prediction intervals over 100 episodes after the last measured performance (from Holt (2004)).
- For each pair $(E_k, d_n)$, algorithm $A_i$ achieves significantly higher trend in performance than $A_j$ in the presence of $d_n$ if $A_i$ has a higher forecast prediction line than $A_j$ with no overlap between the corresponding prediction intervals.

**3. Reporting**

- Experiment Parameters: Report all distribution shift implementations, hyperparameters, code-level optimizations, computational requirements, and framework details.
- Plots: For each environment $E_1 \ldots E_K$, create plots from 2a or 2b (preferably both) for each algorithm $A_1 \ldots A_M$ in the presence of distribution shifts $d_1 \ldots d_N$.
- Analysis: Provide analysis for the plots and derive conclusions.

---

## 5 EXAMPLES OF RL TIME-SERIES ANALYSIS

In our time series evaluation, we model the scenario as deploying the agent(s) out in the intended environment. In this scenario, we have the pretrained agents run over a number of episodes. As mentioned previously, the causal impact procedure can be interpreted as the RL version of car crash tests, where RL agent(s) are deployed in a controlled environment and undergo repeated performance testing. When evaluating the causal impact of the distribution shift, we introduce the shift at some halfway point. We measure the performance of the treatment and control groups, then measure the difference and cumulative impact of the shift. This can be interpreted as the human maintainers measuring robust agent performance and quantifying the losses that can accrue when the agent is deliberately exposed to a distribution shift. In both single and multi-agent settings, we use the causal impact plot template from Brodersen et al. (2015). In the observational case, we can interpret having the agents run over a number of episodes as deploying the agent into the environment each day while recording its performance. The distribution shifts in the observational evaluations happen randomly, so the human maintainers have no control over how the shifts are introduced.

For the observational evaluations, the single-agent and multi-agent cases use different approaches to randomly introducing distribution shift. In the single-agent case, we define a probability threshold

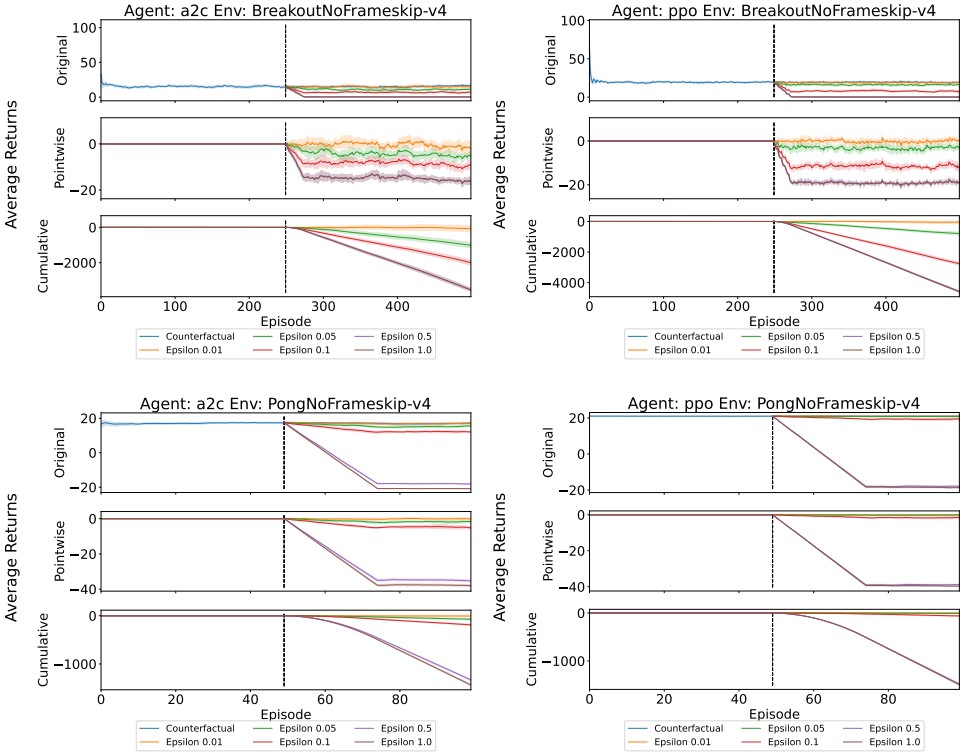

Figure 3: The causal impact plots shown here illustrate the impact of FGSM adversarial attacks on RL agents trained on the Pong Atari game. Each row represents an Atari game. Each column represents a RL algorithm (A2C or PPO). The original plots here show the rolling mean of the rewards over time. The pointwise plots show the difference between the counterfactual performance and the performance when the agent is attacked. The cumulative performance is the summation of the rewards gained or lost over time. As expected, the performance tends to drop as $\epsilon$ increases.

for adversarial attacks. If the random number generator (like `random.random()` in Python) gives a number above the threshold, the Atari game image is attacked. In the multi-agent case, we use a strategy similar to Rahman et al. (2021), where the number of steps an agent is switched out is drawn from a uniform distribution. Figure 4 provides some Atari game examples of observational studies. We use 10 random seeds for both causal and observational time-series evaluation. More information on the plots is provided in the Appendix.

## 5.1 ANALYSIS ON ADVERSARIAL ATTACKS ON A2C AND PPO AGENTS

Here, we measure the performance of RL agents trained on Atari games (Bellemare et al., 2013) in the presence of adversarial attacks. In Figure 3 and the Atari impact graphs in Appendix D, we use pretrained A2C and PPO pretrained agents provided by RL Baselines3 Zoo (RLZoo) (Raffin, 2020) implemented in Stable-Baselines3 (Raffin et al., 2021). For adversarial attacks on the Atari game images, we use the Fast Gradient Sign Method (FGSM) (Goodfellow et al., 2014). In most cases, like in Figure 3, the adversarial attacks work as expected and generally decreases the performance as $\epsilon$ increases. One noticeable pattern is that the agents significantly deviate from the baseline performance when $\epsilon \geq 0.5$. PPO performance is generally equal or worse than A2C after the distribution shift intervention at the halfway point. There are games, like Qbert and Space Invaders, where PPO achieves much worse cumulative returns than A2C when exposed to the highest values of $\epsilon$. Another strange finding was that the RLZoo A2C agent showed non-decreasing (or slightly increasing) cumulative returns in Asteroids and Road Runner. One conclusion that can be drawn from these findings is that the pretrained RLZoo A2C is more robust than PPO against constant adversarial examples.

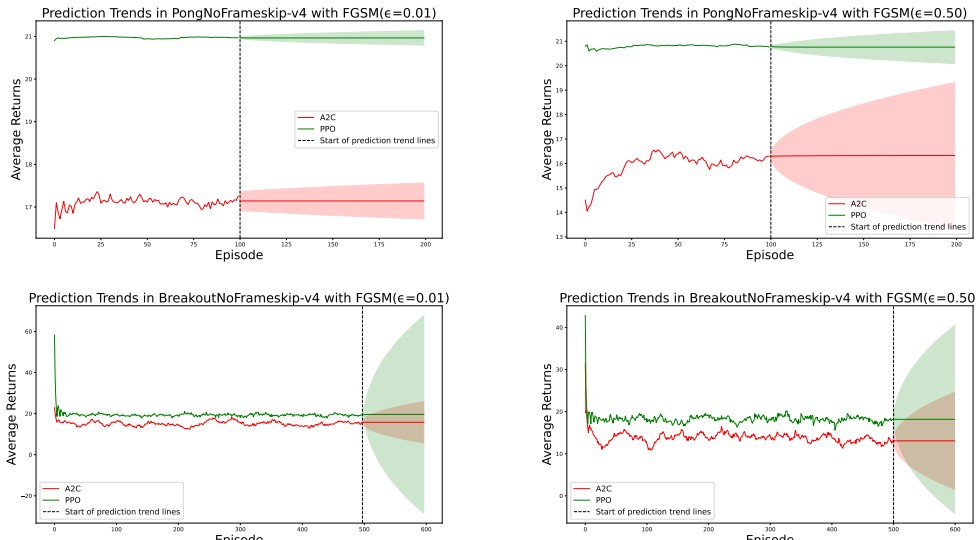

Figure 4: These plots take the rolling mean (window=25) of the observational performance up to a certain time point with some probability of being in the presence of adversarial attacks. After that time point, it shows the time series forecast with 99% prediction intervals. **Top Row:** In the PongNoFrameskip-v4 plots, PPO performs significantly better and has smaller prediction intervals. The plot on the right, however, uses attacks with higher $\epsilon$, where both agents have larger prediction intervals. **Bottom Row:** In the BreakoutNoFrameskip-v4 plots, the prediction intervals overlap. One noticeable difference is that the stronger attacks cause the PPO interval to shrink in size while the mean rewards are slightly less. This decrease in variability accounts for the decrease in the maximum rewards achieved.

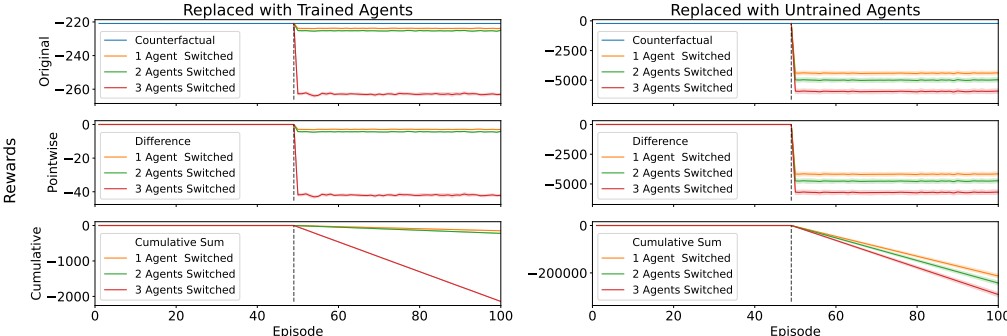

Figure 5: The plots here show the impact of the ad hoc switching of agents in a group of 5 in the PowerGridworld environment. **Left Column:** We replace 1, 2, or 3 agents out of the group of 5 with agents that trained with a different group. While there is little change when only replacing 1 or 2 agents, we see that performance dramatically decreases when 3 agents have been switched out. **Right Column:** We see that just switching out 1 agent in the group with 1 untrained agent causes a significantly large decrease in group performance.

The observational plots, like in Figure 4 and Appendix D, tell a more nuanced story. Except for Pong, where the PPO performance trends are significantly better than A2C, PPO's forecasts trends are higher than A2C's but there is overlap between the prediction intervals. It seems that the PPO agents are able to recover and choose better actions when not observing adversarial examples. This is consistent with the RL Baselines3 Zoo benchmark scores, where PPO scores higher than A2C in every Atari game (without the presence of adversarial attacks). Hence, we conclude that RLZoo PPO agents have trends that perform (not significantly) better in the presence of random attacks, but the impact plots show clear signs of overfitting in some environments. On the other hand, A2C agents tend to be relatively more robust than PPO against adversarial attacks in Atari games.

## 5.2 ANALYSIS ON AGENT SWITCHING IN POWERGRIDWORLD

In the multi-agent case (Figure 5), we measure the performance of a cooperative group of 5 decentralized PPO agents in the presence of ad hoc agent switching. The multi-agent environment we use is a framework for power-systems-focused simulations called PowerGridworld (Biagioni et al., 2022). As seen on the impact plots in Figure 5, we investigate what happens when 1 to 3 agents in the group are switched out with the corresponding number of outside pretrained or untrained agents. The outside pretrained agents were trained as a separate group of 5 agents that achieve a score similar to the original group. When switching out the agents with the outside group of pretrained agents, there was a slight impact when switching out 1 or 2 agents. The performance dramatically decreased when 3 agents were switched out, which is a majority of the agents. Replacing one agent of the group with one untrained agent, as can be seen in Figure 5, is much worse than switching out 3 agents with outside pretrained agents. The observational PowerGridworld plots in Appendix D provide complementary results. Hence, in this PowerGridworld environment, we can conclude that (i) switching out and replacing the majority of agents (including pretrained) can significantly diminish performance, and (ii) replacing even one trained agent with an untrained agent can be worse than switching out the majority with outside trained agents. When switching the original agents with similarly-trained outside agents, it seems that group performance is robust when only switching out a minority of the agents, but is significantly less robust when a majority of agents is switched out.

## 6 CONCLUSION

In this work, we show that relying on point estimates, even alternatives that are more reliable and less biased than what is typically used in practice, are limited in their ability to evaluate RL performance in the presence of distribution shifts. In particular, we argue that the methodology proposed here provides more of a necessary emphasis on test-time evaluation, and can assess both single and multi-agent performance. If, or when, we choose to deploy RL agents into the real world, how such agents perform after training will matter more than during training. The decline in performance during ad hoc agent switching in the PowerGridworld shows what could happen if we need to start replacing intelligent agents to ensure energy usage is minimized. Such experiments reveal a workflow that will likely be closer to representing real-world safety checks and maintenance of AI systems. Our methodology might also be useful for investigating general single or multi-agent RL behavior.

There is still work to accomplish in ensuring RL research is reproducible. A weakness of our approach is that it does not account for non-deterministic environments. We also did not investigate if more complex time series models would be more appropriate in certain scenarios would be an obvious next step. Time series clustering and motifs (Mueen et al., 2009; Imani et al., 2021) can be used to better understand how different types of distribution shift can affect RL performance. One could also investigate if time series ensemble models (Wichard & Ogorzalek, 2004) provide better forecasts than simpler models. Finding answers to these shortcomings can be investigated in future research.

Nations, like the United States (White House, 2022), have signed legislation to invest in modernization of infrastructure and communities to meet the growing challenges of the 21st century (e.g., cybersecurity and climate change). RL is advancing the automation of complex decision making in real-world infrastructure systems (e.g., energy and transportation). Like other areas in AI, the role of RL in critical infrastructure and other high-consequence applications requires robust, reliable, repeatable, and standardized evaluation protocols (European Union, 2021; Biden Jr, 2023). The methodology we propose here, inspired by the NHTSA ratings standards, is a start toward a general, standardized protocol for evaluating pretrained RL performance over time.

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
