APPENDIX

## A Methods

We provide the following definition of RL agent time-series performance measurements, similar to how performance is measured in RL training research, that will be relevant in the next subsection:

**1 Definition.** Let $X^A(t)$ the time series performance for $t = 1 \ldots N$, where $A$ is an RL agent in a simulated deterministic environment $\mathcal{M}$. For each $X^A(t)$, there is an associated sample of performance measurements $x^A(t) = (x^{A,1}(t), \ldots, x^{A,M}(t))$. Here, the $i$ in $x^{A,i}(t)$ is a random seed $i \in \mathfrak{S}$, where $\mathfrak{S}$ is a finite set of $M$ random seeds. Hence, we define:

$$X^A(t) = \mathbb{E}[x^{A,i}(t)] = \frac{\sum_{i \in \mathfrak{S}} x^{A,i}(t)}{M} \tag{1}$$

which is the expected performance[1] at time $t$ over all the random seeds in $\mathfrak{S}$.

### A.1 Comparison of Simple Time Series Forecasting Trends

To compare RL algorithms at test time, we compare the performance forecasts of each RL agent. This recommendation is inspired by the use of time complexity or running time to measure the efficiency of an algorithm (Cormen et al., 2022). As illustrated in Figure 1, looking at performance trends of RL agents is an informative way to observe overfitting during test time. Here, we use simple time series forecasts over more complex time series models that might better predict the performance trend. The reason is that one cannot make too many strong, general assumptions (e.g., seasonality, how many past values to use for autoregression) on the time series trend of agent performance in all environments. For test-time distribution shift, we only assume the trend will be pessimistic and not increase. This is reasonable if we assume the ideal case is when performance never decreases (like Agent 1 in Figure 1). We default to using Holt's linear damped trend method (Gardner Jr & McKenzie, 1985; Holt, 2004; Hyndman & Athanasopoulos, 2018) to model the trend of agent performance when in the presence of distribution shift. The damping parameter discourages constant increasing trends, which will likely give more accurate forecasts than Holt's method without damping. Researchers evaluating their own experiments, however, may find that more complex models would be more appropriate for their own studies if they see performance trends that warrant such assumptions.

Here, we define Holt's linear damped trend method (Hyndman & Athanasopoulos, 2018):
Let $\hat{y}_{t+h|t}$ be the short-hand estimate for $y_{t+h}$ based on data $y_1 \ldots y_t$. The term $\ell_t$ is the estimate of the level of the series at time $t$ with a smoothing parameter $\alpha$ such that $0 \le \alpha \le 1$. The term $b_t$ is the estimate of the trend (slope) of the series at time $t$ with a smoothing parameter $\beta^*$ such that $0 \le \beta^* \le 1$. Let $\phi$ be the damping parameter such that $0 < \phi < 1$. Then the trend method is represented as the forecast equation and smoothing equations below:

$$\hat{y}_{t+h|t} = \ell_t + (\phi + \phi^2 + \cdots + \phi^h)b_t$$
$$\ell_t = \alpha y_t + (1 - \alpha)(\ell_{t-1} + \phi b_{t-1})$$
$$b_t = \beta^*(\ell_t - \ell_{t-1}) + (1 - \beta^*)\phi b_{t-1}$$

### A.2 Prediction Intervals over Future Performance

Along with time series forecasts, we recommend prediction intervals over future average returns (Hyndman & Athanasopoulos, 2018). The combination of simple time series forecasts and prediction intervals map out the range of average returns over time. As stated previously, point estimates with confidence intervals are not enough to capture the range of returns over time. Prediction intervals act as a compliment to the time series forecasting by visualizing the uncertainty of the possible future average returns. When applying both of these methods, we can specify the most robust RL algorithm as the trend with the most optimistic forecast on performance and the smallest prediction interval. Here, we assume there exists distribution shifts but do not necessarily know when or where.

---

[1]We use expected performance just as a simple default. For example, in accordance with Agarwal et al. (2021), one could replace expected performance with the interquartile mean.

## A.3 DIFFERENCE-IN-DIFFERENCES ANALYSIS FOR RL PERFORMANCE

Difference-in-differences (DiD) measures the causal effect between a treatment and control group, where the treatment group is exposed to an intervention at a certain point in time (Cunningham, 2021; Huntington-Klein, 2021). Intuitively, it represents how much more the treatment group was affected by the intervention at some time $t$ compared to the change of the unaffected control group at the same time $t$. Since we are dealing with time series, we will adhere to the DiD formulation in Moraffah et al. (2021):

**2 Definition.** If $t < T$ and $t > T$ denote the pre- and post- treatment periods, respectively, then we can calculate the DiD measure using the average treatment effect metric over a time series $X(t)$ as follows:

$$DiD = \{\mathbb{E}[X(t > T)|G = 1] - \mathbb{E}[X(t < T)|G = 1]\} - \{\mathbb{E}[X(t > T)|G = 0] - \mathbb{E}[X(t < T)|G = 0]\} \tag{2}$$

where $G$ indicates the treatment group ($G = 1$) and the control group ($G = 0$).

Any methodology that relies on DiD must satisfy the **parallel trends assumption**, which says that *if no treatment had occurred, the difference between the treated group and the untreated group would have stayed the same in the post-treatment period as it was in the pre-treatment period* (Huntington-Klein, 2021). RL agents in deterministic environments with fixed seeds trivially satisfy this because the agent will take the same action at each state and receive the same reward regardless of how many times the evaluation is repeated.

From the RL fixed seed assumption, DiD simplifies to the equation below when evaluating RL performance:

$$DiD = \mathbb{E}[X^A(t > T)|G = 1] - \mathbb{E}[X^A(t > T)|G = 0] \tag{3}$$

where $A$ is the RL agent being evaluated. This follows from the pre-treatment averages canceling each other out. Hence, we only need to measure the post-treatment effect of RL performance. Using the RL fixed seed assumption, DiD becomes the following:

$$DiD = \mathbb{E}[X^A(t > T)|G = 1] - \mathbb{E}[X^A_{U=0}(t > T)|G = 1] \tag{4}$$

Equations 3 and 4 say the following: Measuring the DiD effect is equivalent to measuring the average time series post-treatment effect of agent $A$ between the treatment and control group. If the agents in both the treatment and control groups have the same fixed random seeds, we can interpret the performance measurements of the control group as the counterfactual of the treatment group as if the treatment group was never exposed to the distribution shift intervention. Hence, we have shown that the RL fixed seed assumption justifies causal inference in our time series analysis.

## B  FURTHER NOTES ON THE RL FIXED SEED ASSUMPTION

Reproducibility can be difficult to achieve when running on a GPU. As described in PyTorch's webpage on reproducibility, even identical seeds might not provide reproducible results. Some reasons include nondeterministic algorithms that improve performance and the use of different hardware can affect the selection of such algorithms. To control these sources of randomness in our experiments, we adhere to the reproducibility suggestions provided on the webpage.

## C  FURTHER NOTES ON ENVIRONMENTS USED IN EVALUATIONS

All plots use 10 random seeds. In multi-agent settings, each agent shares the same seed during an evaluation run. For example, if our seed numbers are 1 and 42, then the first evaluation run sets

### C.1  ATARI GAMES

The games of focus are a subset of Atari games AsteroidsNoFrameskip-v4, BeamRiderNoFrameskip-v4, BreakoutNoFrameskip-v4, MsPacmanNoFrameskip-v4, PongNoFrameskip-v4,

QbertNoFrameskip-v4, RoadRunnerNoFrameskip-v4, SeaquestNoFrameskip-v4, and SpaceInvadersNoFrameskip-v4. The agents we evaluate are pretrained agents from RL Baselines3 Zoo (Raffin, 2020) that are available on SB3's Huggingface repository of models. The adversarial attacks (FGSM) were implemented in torchattacks (Kim, 2020).

We evaluate each agent over 100 games per Atari game. In the causal impact plots, the attacker intervenes at the 50th game. Here, since game scores accumulate across lives, we define episodes as the life of the player. For example, Pong only gives one life, which gives us a total of 100 episodes. In Qbert, the player is given 4 lives. Hence, the attacker intervenes at episode 200 because (50 games) * (4 episodes/game) = 200 episodes (or lives). This reasoning is also why there are 400 episodes in the Qbert experiments. This strategy deliberately avoids the noisy artifacts that can emerge from the time series data when accumulated performance suddenly drops when each game ends.

### C.2 POWERGRIDWORLD

PowerGridworld is a modular, customizable framework for building power systems environments to train RL agents. Because of this, we use an environment provided in one of the example scripts. The class name is called `CoordinatedMultiBuildingControlEnv`, which is a multi-agent coordination environment. In addition to the original agent-level reward, grid-level reward/penalty and system-level constraint(s) are considered. In particular, we consider the voltage constraints: agents need to coordinate so the common bus voltage is within the ANSI C.84.1 limit. If the constraints are not satisfied, the voltage violation penalty will be shared by all agents. The agents in this scenario are minimal implementations (Barhate, 2021) of PPO.

### C.3 TIME SERIES TOOLS

Trends and prediction intervals were implemented in the Python package sktime (Löning et al., 2022). Other Python visualization tools include Matplotlib (Hunter, 2007) and Seaborn (Waskom, 2021).

## D MORE PLOTS

### D.1 ATARI GAME CAUSAL IMPACT PLOTS

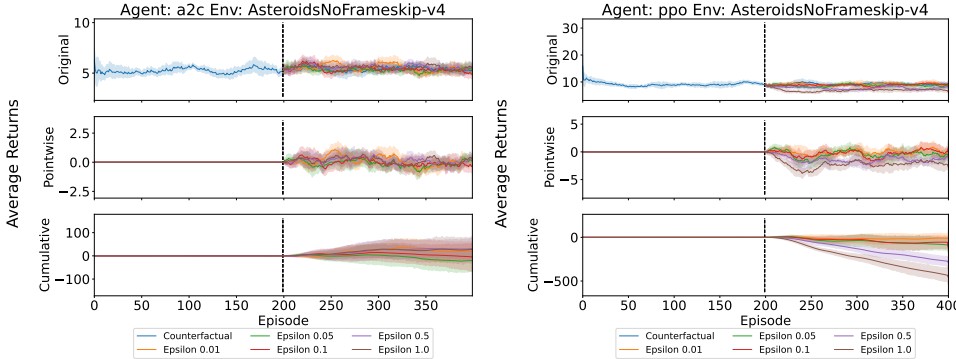

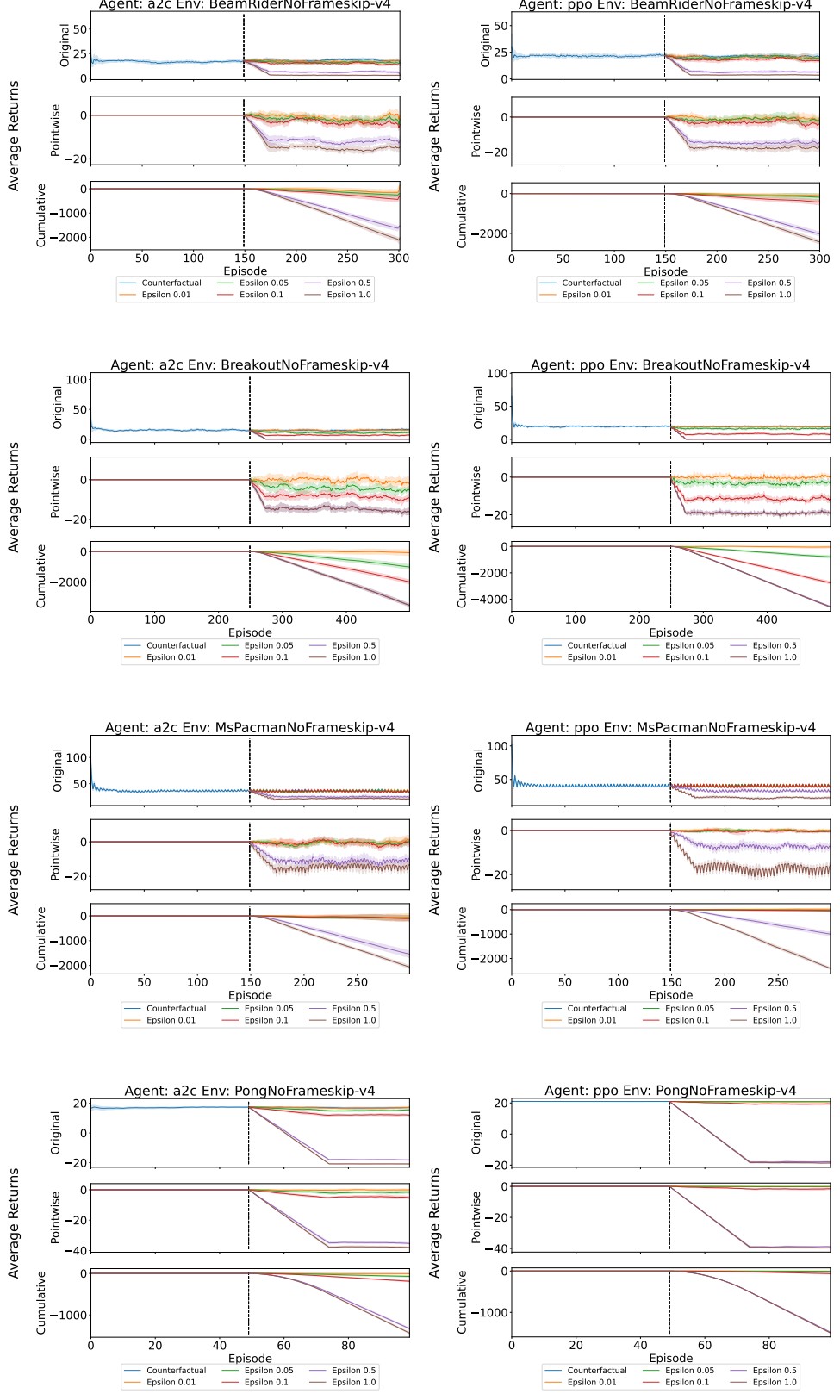

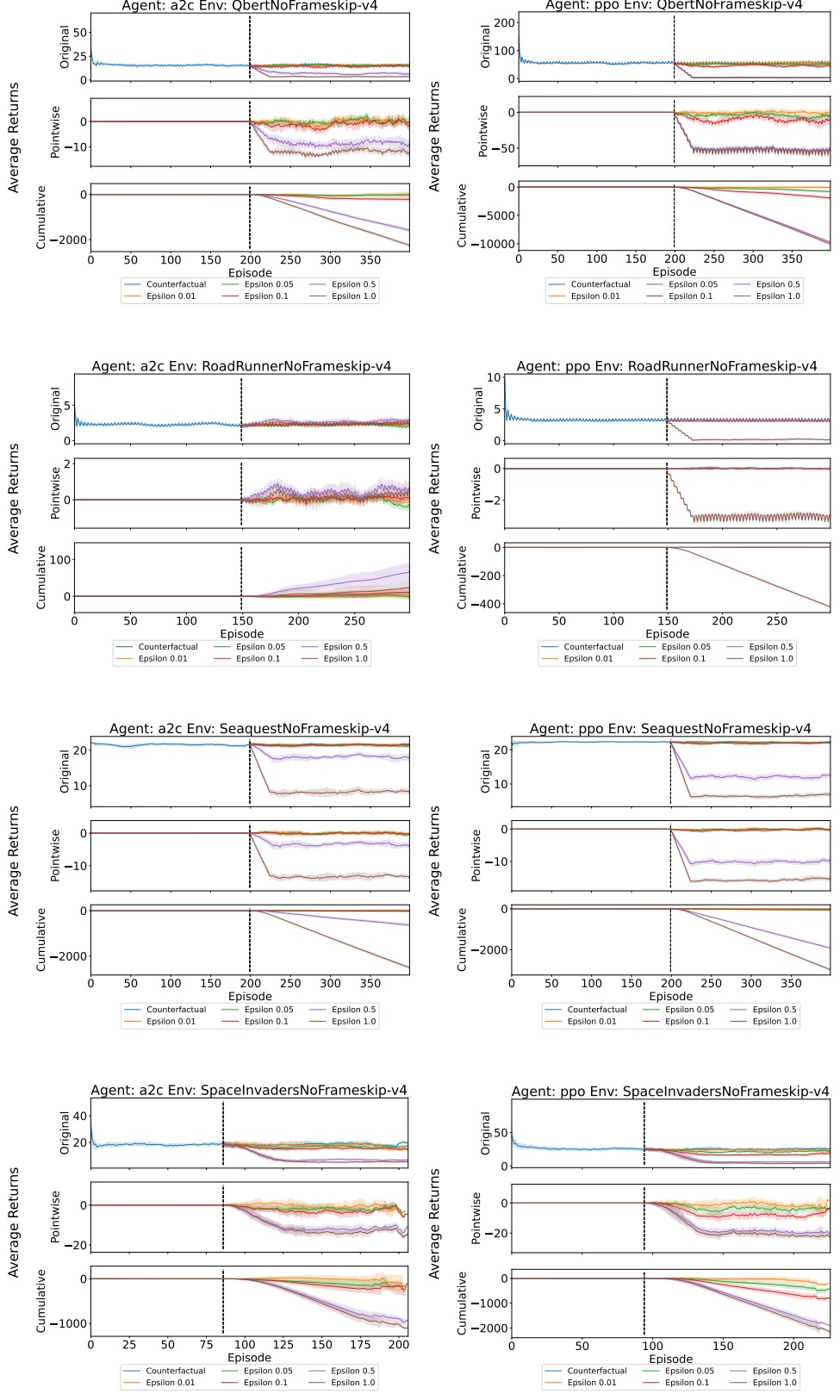

### D.2 OBSERVATIONAL PLOTS

All observational plots show forecasts with prediction intervals 100 episodes after the last measurement.

### D.2.1 POWERGRIDWORLD

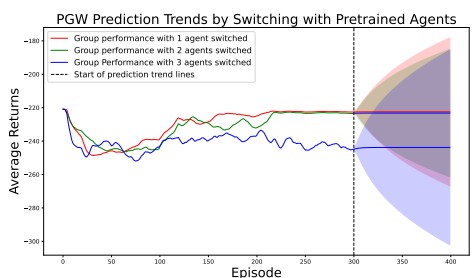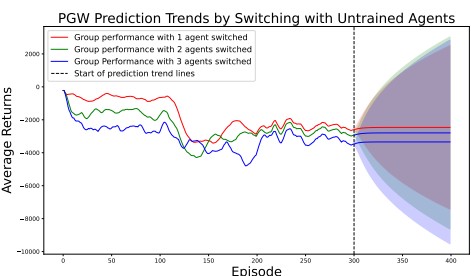

Figure 1: PowerGridworld observational plots. **Left:** Comparing random switching with pretrained agents at each episode. **Right:** Comparing random switching with untrained agented at each episode.

### D.2.2 ATARI GAMES

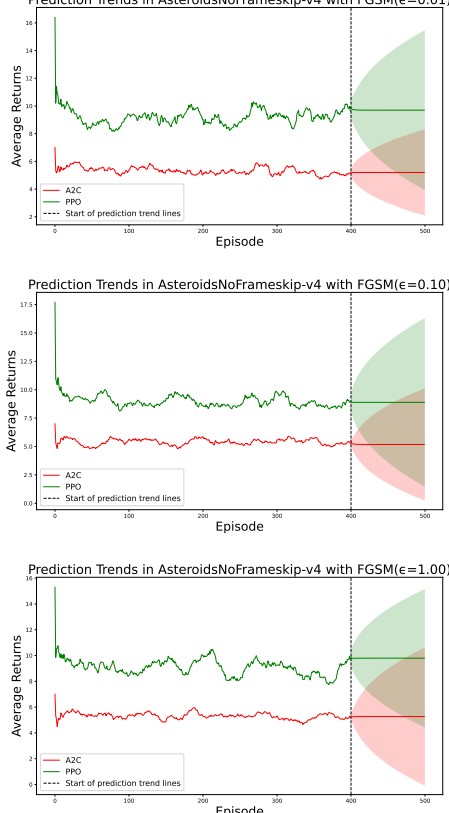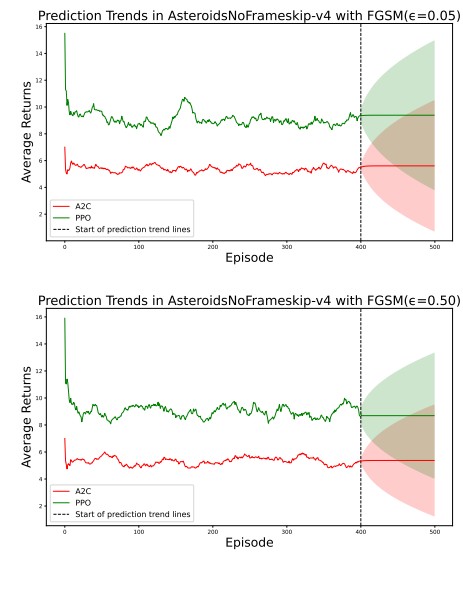

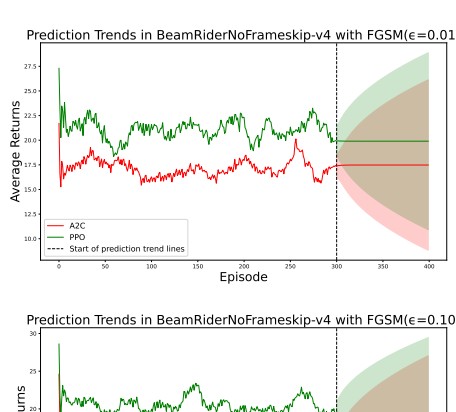

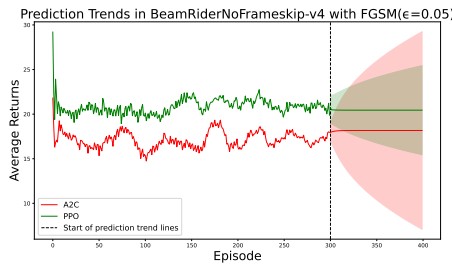

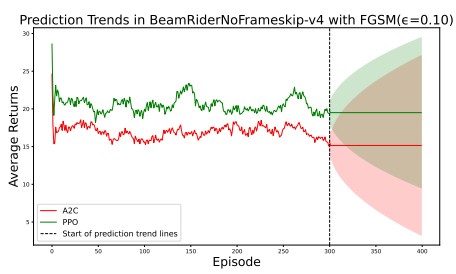

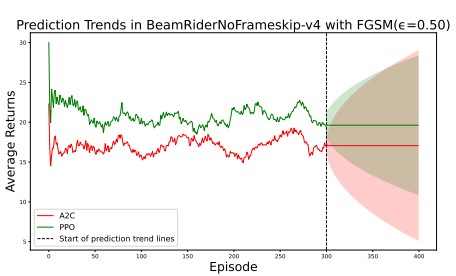

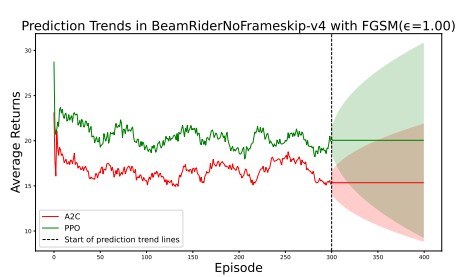

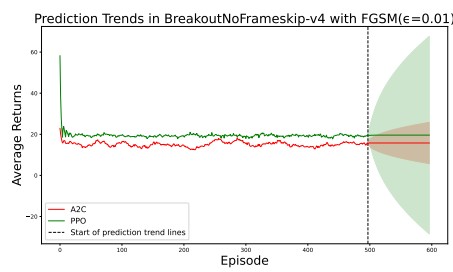

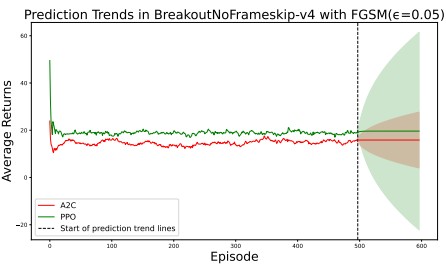

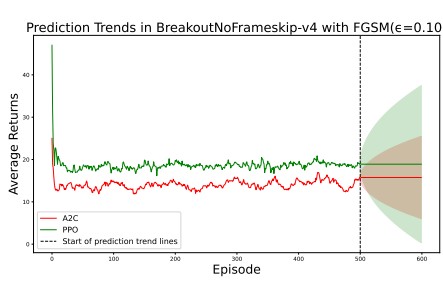

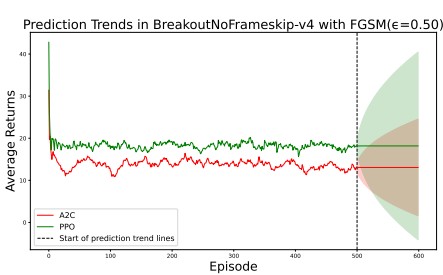

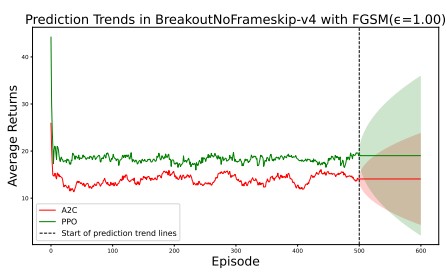

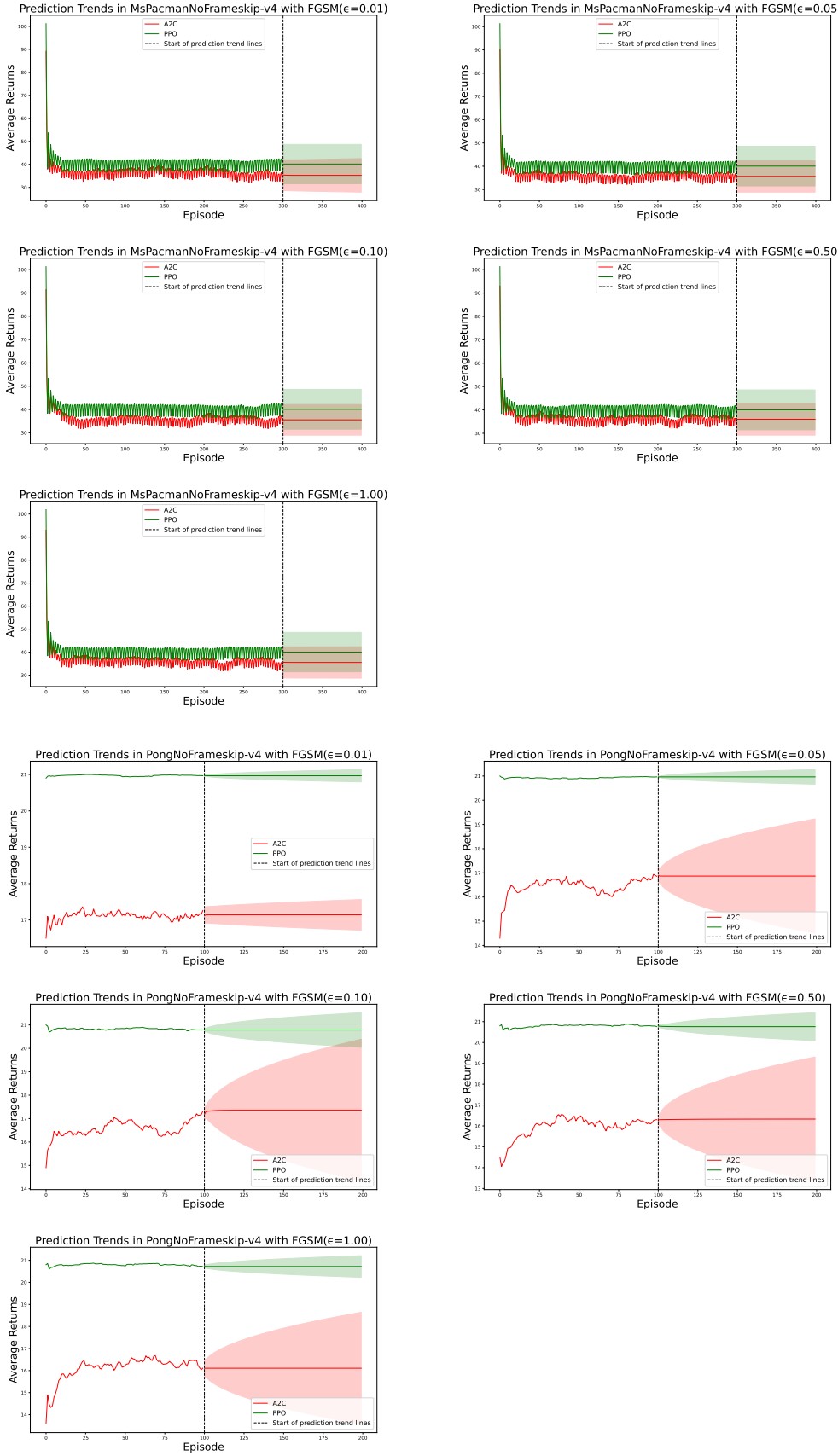

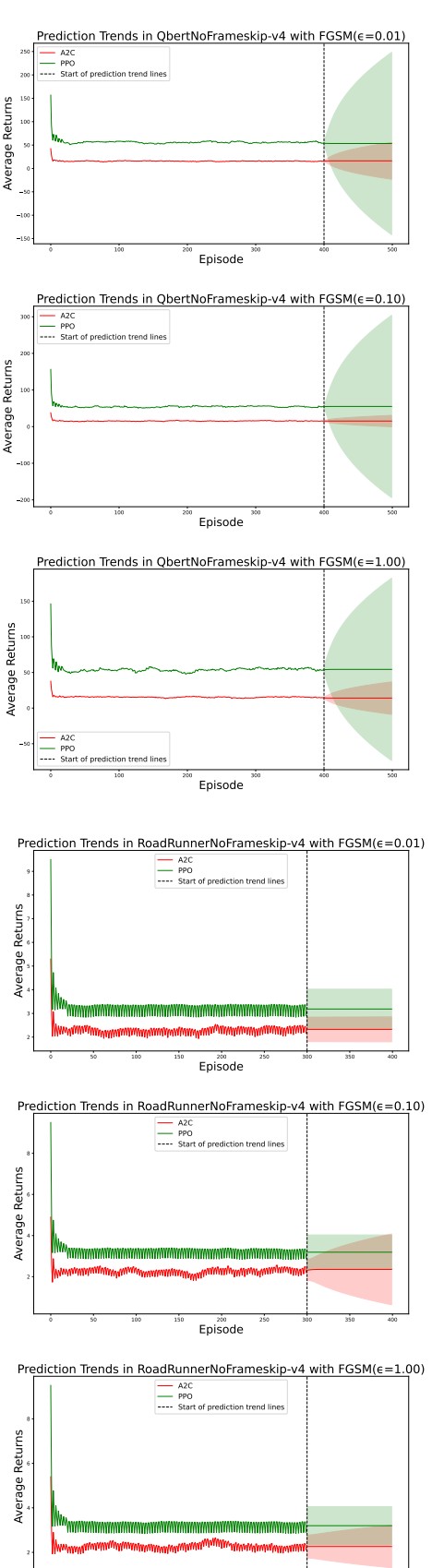
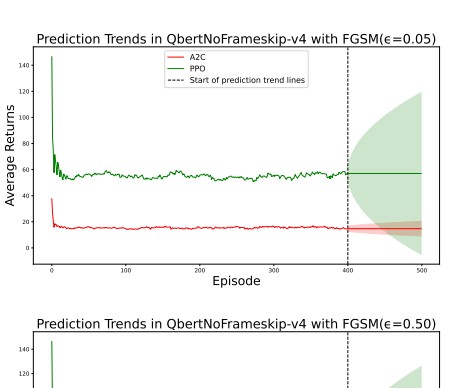
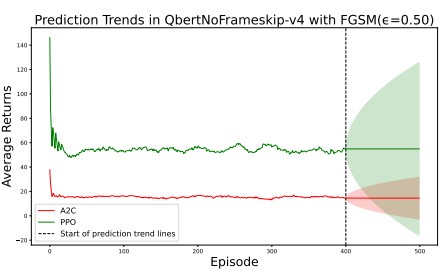
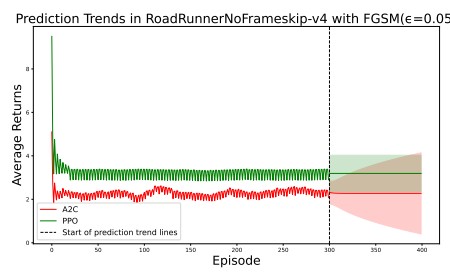
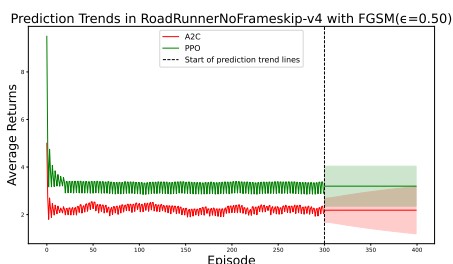

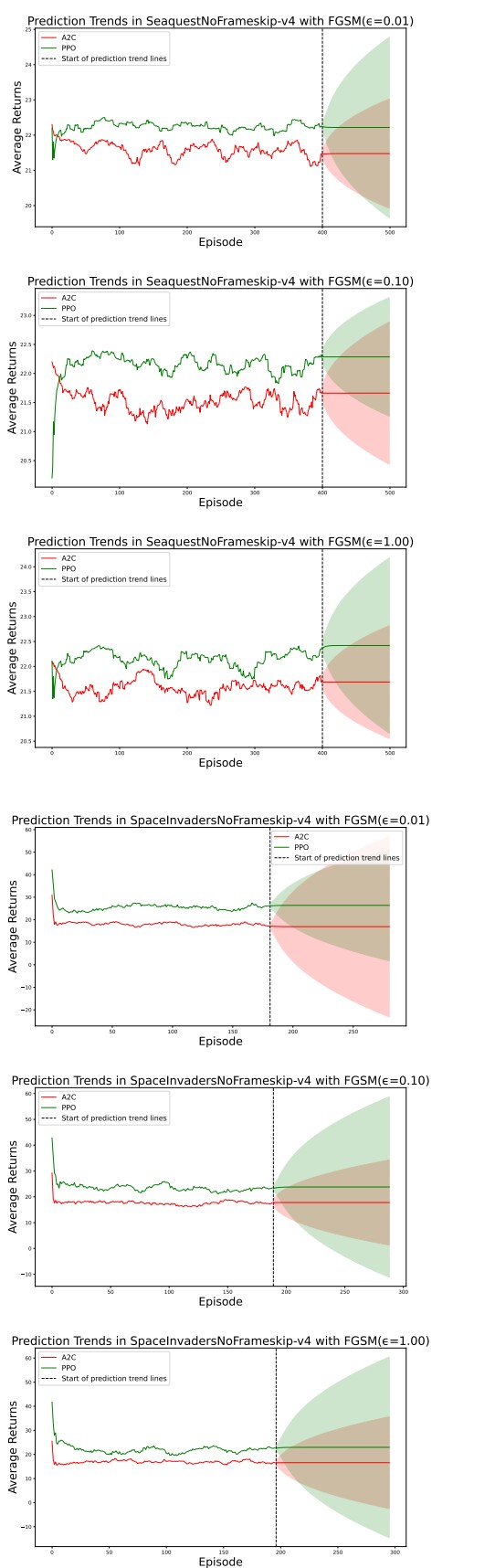

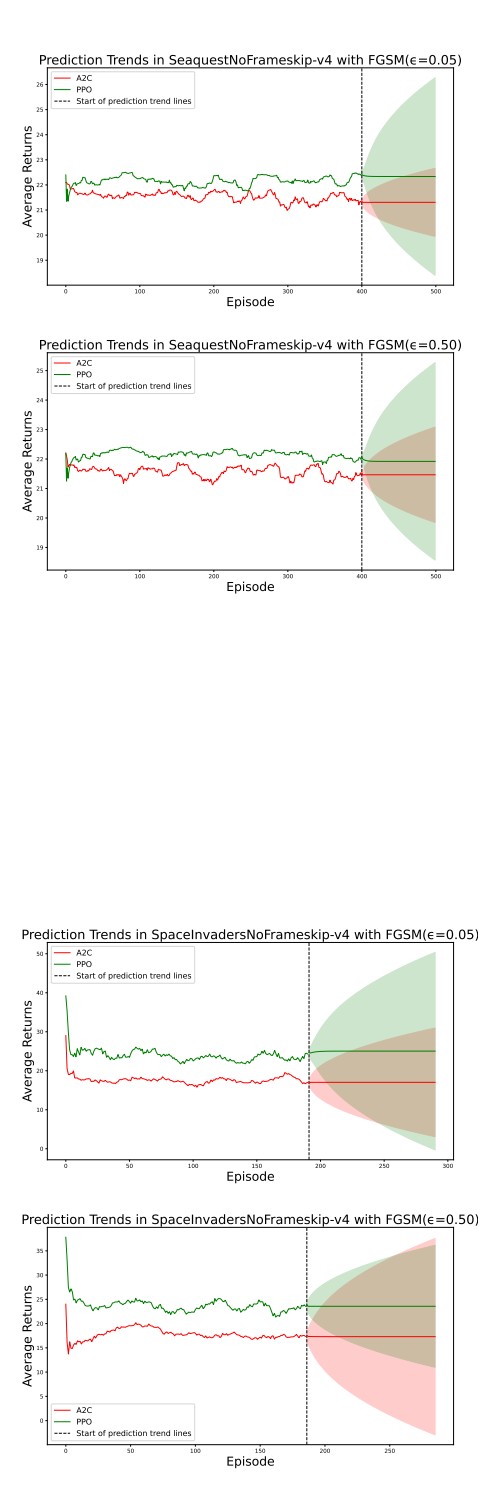

# E HYPERPARAMETERS FOR POWERGRIDWORLD PPO TRAINING

Number of Training Episodes: 2,000,000
Maximum Number of Steps per Episode: 1000
Starting Standard Deviation for Action Distribution: 1.0
Standard Deviation for Action Distribution Decay Rate: 0.05
Minimum Standard Deviation for Action Distribution: 0.1
Standard Deviation Decay Frequency in time steps: 250,000

Policy: 2-layer Multilayer Perceptron
Hidden Units: 128 per layer
Policy is updated every 2000 time steps. During that time step, the policy updated 5 epochs.

Clip parameter: 0.2
Discount factor $\gamma$: 0.99

Actor Learning Rate: 0.0003
Critic Learning Rate: 0.001

Main group random seed: 0
Outside group random seed: 1