# OpenReview forum: "Assessing the Impact of Distribution Shift on Reinforcement Learning Performance"
_ICLR.cc/2024/Conference — Submitted to ICLR 2024_

### Official Review · Reviewer_okG3 · 2023-10-31

**Soundness:** 3 good
**Presentation:** 3 good
**Contribution:** 2 fair
**Rating:** 6
**Confidence:** 3

**Summary:**

Reinforcement learning (RL) contains unique challenges in fixing its reproducibility problem. Current displays of evaluation take attention away from other important factors such as model overfitting and experimental design. RL researchers have developed various reliability evaluation metrics to understand the strengths and weaknesses of each RL algorithm, but these metrics do not take out-of-distribution observations into account. The authors propose time series analysis tools to measure model robustness under the presence of distribution shift. They apply these analytical tools in both single-agent and multi-agent environments to show the effect of introducing distribution shifts during test time.

**Strengths:**

I appreciate the plot showing the flaw of solely relying on point estimates to perform model evaluation.

**Weaknesses:**

The authors don't list potential weaknesses of their current method in the main text.

**Questions:**

N/A

---

> ### Author Response · Authors · 2023-11-18
> **We thank the reviewer for the feedback**
>
> > **The authors don't list potential weaknesses of their current method in the main text.**
>
> Thank you for pointing that out. The second paragraph in section 6 of the improved version now mentions some possible weaknesses of our methodology.
>
> We also invite reviewer okG3 to read the improved version of the main article and appendix. There are some significant changes that we believe enhance the quality of the paper.

---

> > ### Comment · Reviewer_okG3 · 2023-12-02
> > **Response to Authors**
> >
> > Thank you for your response. I am satisfied with the authors' edits and as a result, I have increased my score on the paper.

---

### Official Review · Reviewer_8p4f · 2023-11-01

**Soundness:** 2 fair
**Presentation:** 2 fair
**Contribution:** 1 poor
**Rating:** 3
**Confidence:** 3

**Summary:**

This paper studies the evaluation of RL algorithms against distribution shifts.
In particular, it proposes the usage of evaluation techniques from the time series literature to take into account changes in the environment.

**Strengths:**

- The paper is written in an intuitive way, which would help the adoption of the proposed methodology by the RL community.

- The evaluation of RL algorithms is certainly an important issue

**Weaknesses:**

- It did not come across to me what is the precise problem the paper is trying to solve. The problem formulation is somewhat vague and relies on artificial examples that make it difficult to connect it with the application of RL algorithms.

- The methodology proposed is somewhat scattered. It is unclear how this evaluation methodology would be applied. Perhaps the paper could include a pseudo-code or a flowchart to ground all the steps of the proposed methodology.

- Although the paper provides several examples, it does not provide a proper interpretation of the results. For example, in Figure 5, which agent is more robust? In Figure 3, what is the conclusion regarding the performance of A2C and PPO? Should we favor one of them in practice?

- The related work section only lists the contribution of related papers but does not provide a description of how this paper distinguishes from those.

**Questions:**

1. Could you describe more formally the assumption that "the trained agents achieved a clear trend in performance"? In particular, how is this related to the convergence of an RL agent? Does it mean the agent has reached an optimal performance? Furthermore, as the assumption does not consider a distribution shift, does it mean the agent continues to update its behavior?
2. Could you comment on the connections from this methodology with non-stationary MDPs?
3. Considering the change in the dynamics of the environment, I think we cannot always conclude that the change in performance is due to the agent's behavior. For instance, in some cases, although the dynamics change, the optimal policy may remain the same. Could this methodology help identify if the agent is underperforming?

---

> ### Author Response · Authors · 2023-11-18
> **We thank the reviewer for their valuable feedback**
>
> > **It did not come across to me what is the precise problem the paper is trying to solve. The problem formulation is somewhat vague and relies on artificial examples that make it difficult to connect it with the application of RL algorithms.**
>
> Sorry for not making this clearer. Accounting for distribution shift has not been previously investigated in reliable RL. This is a more realistic setting for agents that will be deployed in the real world. Hence, we wanted to provide a methodology for this under-researched problem that measures RL performance under distribution shift using time series. Past RL reliability papers rely on point estimates alone. Time series analysis is also needed check if a well-trained agent continues to perform well over a period of time. For example, you would want a methodology in controlled settings that shows how a RL-trained self-driving car performs in the presence of many distribution shifts over a long-enough period of time before releasing it out to the public. The new protocol at the end of section 4 should clarify our proposed methodology. The new analyses in section 5 show that our methodology does not rely on artificial examples.
>
> > **It is unclear how this evaluation methodology would be applied. Perhaps the paper could include a pseudo-code or a flowchart to ground all the steps of the proposed methodology.**
>
> Thank you for pointing this out. To fix this, we included a clear step-by-step protocol at the end of section 4.
>
> > **In Figure 5, which agent is more robust? In Figure 3, what is the conclusion regarding the performance of A2C and PPO? Should we favor one of them in practice?**
>
> Figure 5 alone is just meant to show what it looks like when there is a significant difference is and what it looks like when the difference is not significant in the observational experiments. As mentioned, section 5 shows some analyses regarding what conclusions to draw from our methodology. We hope the new Atari analysis in section 5, which uses Figure 5 as part of the story, gives a sufficient interpretation of the results in Atari games.
>
> >**The related work section only lists the contribution of related papers but does not provide a description of how this paper distinguishes from those.**
>
> Thank you for pointing this out. We included a few more sentences at the end of the last paragraph of section 2 of the improved version to distinguish ourselves from past RL reliability work.
>
> > **Could you describe more formally the assumption that "the trained agents achieved a clear trend in performance"? In particular, how is this related to the convergence of an RL agent? Does it mean the agent has reached an optimal performance? Furthermore, as the assumption does not consider a distribution shift, does it mean the agent continues to update its behavior?**
>
> Thank you for catching that. What we should have said was "a flat (slope = 0) trend in raw performance." We updated the article to fix this issue. Here, we just assume some baseline performance (not necessarily optimal) for the causal impact plots. We do not assume the agent updates its behavior. That might be interesting for future research to see how agents with adaptive learning can fare under this methodology.
>
> > **Could you comment on the connections from this methodology with non-stationary MDPs?**
>
> Sure! With some modifications, our methodology should still hold if the parallel trends assumption holds for the treatment and control groups. This would require verification that both groups show parallel trends in the current environment, which I don't expect to hold in all cases. If this line of thinking were to continue, I expect we would need to use other sophisticated methods of causal inference to draw conclusions about agents in non-stationary environments. One drawback of going outside of stationary MDPs is the possible need to use environment-specific knowledge (or covariates besides rewards), which we deliberately avoid in this paper to keep it simple and general. This is out of the scope of this paper, but interesting nonetheless.
>
> > **Considering the change in the dynamics of the environment, I think we cannot always conclude that the change in performance is due to the agent's behavior. For instance, in some cases, although the dynamics change, the optimal policy may remain the same. Could this methodology help identify if the agent is underperforming?**
>
> We do not claim our methodology necessarily assigns blame between agent and environment for the decrease in agent performance. Yes, it could be that the agent was able to reach optimal behavior in the training environment, but that does not necessarily mean it will still perform optimally when the test environment is different in some way. Our framework can identify if the agent is underperforming if you know what distribution shift to use during the causal impact experiments. Hopefully, things will become clearer if you read the protocol in section 4.

---

### Official Review · Reviewer_mUDk · 2023-11-05

**Soundness:** 3 good
**Presentation:** 2 fair
**Contribution:** 1 poor
**Rating:** 5
**Confidence:** 3

**Summary:**

This paper proposes a set of evaluation methods that measure the robustness of Reinforcement Learning (RL) algorithms under distribution shifts. The paper argues to account for performance over time while the agent is acting in its environment. The authors recommend time series analysis as a method of observational RL evaluation and show that the unique properties of RL and simulated dynamic environments supports their additional assumptions needed to measure the causal impact in their experimental evaluation.

**Strengths:**

1. Detailed background of the various causal inference topics is provided. Even though most of the information can be moved to the appendix, this amount of information makes the paper easy to read for someone new to the field.

**Weaknesses:**

1. Lack of novelty: The paper does not have a novel contribution. The idea of using simulators to perform interventional analysis is not new [1-5].

Though they talk about the focus in the paper being on "adversarial attacks on images (Atari game observations) and agent switching in multi-agent environments", there is nothing that is specific to the adversarial nature of the distribution shifts. The paper as is will be applicable if distribution shifts happen due to any other factor.

2. Verbose description: The setup is unnecessary and made complex to justify the simple idea of using a simulator to perform interventions. Simple things like an average of sampled data is dedicated a definition and equation (eq. 1), which I think is not needed. Only Section 4.2 is something that talks about a new approach, everything before that is motivation or background.

3. Insufficient experimental analysis: Experimental evaluation is not sufficient. Figures 3 and 4 are not analyzed in detail, and the inferences from these experiments are not explained properly. Even in the appendix, only the plots are added without any analysis.

[1] Lee et al., SCALE: Causal Learning and Discovery of Robot Manipulation Skills using Simulation. CoRL 2023.

[2] Verma et al., Learning Causal Models of Autonomous Agents using Interventions. KEPS 2021.

[3] Lee et al., Causal Reasoning in Simulation for Structure and Transfer Learning of Robot Manipulation Policies. ICRA 2021.

[4] Ahmed et al., CausalWorld: A Robotic Manipulation Benchmark for Causal Structure and Transfer Learning. ICLR 2021.

[5] Verma et al., Autonomous Capability Assessment of Black-Box Sequential Decision-Making Systems. KEPS 2023.

**Questions:**

1. How would you comment on the related literature showing that simulators can be used for interventions? How is your idea new compared to them? My failure to understand this is the biggest reason for my score. Maybe I am missing something, and I would appreciate it if you could comment on it.

2. Is the choice of adversarial nature of distribution shift important to the ideas presented in the paper? If we perform an interventional analysis on this paper and make the distribution shifts non-adversarial (something simply changed in the environment), then would the analysis you present still hold? If not, why?

---

> ### Author Response · Authors · 2023-11-18
> **We thank the reviewer for their valuable feedback**
>
> > **How would you comment on the related literature showing that simulators can be used for interventions? How is your idea new compared to them? My failure to understand this is the biggest reason for my score. Maybe I am missing something, and I would appreciate it if you could comment on it.**
>
> We apologize for the confusion. Our paper is a RL reliability/robustness paper. We are following past RL reliability papers like [Measuring the Reliability of Reinforcement Learning Algorithms](https://openreview.net/forum?id=SJlpYJBKvH) and [Deep Reinforcement Learning at the Edge of the Statistical Precipice](https://openreview.net/forum?id=uqv8-U4lKBe). The strength and originality of these papers is not in the metrics themselves, but the proposed protocols that ensure reproducible and reliable RL. We contribute a methodology that includes counterfactual time series analysis, using the impact visualization strategy from Brodersen et al. (2015), as a way to measure the impact of distribution shift after training.  For clarity, we explicitly state in the first paragraph of section 2 of our improved version of the article that it is not a causal machine learning paper and does not claim to be the first to use simulators for interventions.
>
>
>
> > **Verbose description: The setup is unnecessary and made complex to justify the simple idea of using a simulator to perform interventions. Simple things like an average of sampled data is dedicated a definition and equation (eq. 1), which I think is not needed. Only Section 4.2 is something that talks about a new approach, everything before that is motivation or background.**
>
> Thank you for the suggestion. In the updated article, we moved eq. 1 and section 4.1 to the appendix.
>
>
> > **Insufficient experimental analysis: Experimental evaluation is not sufficient. Figures 3 and 4 are not analyzed in detail, and the inferences from these experiments are not explained properly. Even in the appendix, only the plots are added without any analysis.**
>
> Thank you for pointing this out. In section 5, we included more in-depth analyses in section 5. In the Atari games, we conclude that RL Baselines3 Zoo PPO agents tend to perform better but the impact plots show clear signs of overfitting in some environments. On the other hand, A2C agents tend to be relatively more robust against adversarial attacks in Atari games.
>
>
> > **Is the choice of adversarial nature of distribution shift important to the ideas presented in the paper? If we perform an interventional analysis on this paper and make the distribution shifts non-adversarial (something simply changed in the environment), then would the analysis you present still hold? If not, why?**
>
> Sorry about the confusion. No, the choice of using adversarial examples was mainly because it is one of the more well-known types of distribution shift. It was not our intent to focus exclusively on adversarial distribution shifts. In ad hoc agent switching, our original motivation was benign switching of robots that work in a group (e.g. switching teammates in robot soccer/football). We will explicitly state that any distribution shift can be used in our methodology in the second paragraph of section 2 in the updated article.

---

> > ### Comment · Reviewer_mUDk · 2023-11-23
> >
> > I thank the authors for their detailed response. Your response clarifies some aspects, but I still feel the paper itself is not clear enough and can improve with another pass of rewriting. For instance, I gave an example of verbose writing, but it is still verbose in many places, where the same sentence or concept is repeated in multiple places. I do believe the direction is promising, and the modified version is better than the original version. I will increase my score to reflect the changes and better clarity.

---

### Author Response · Authors · 2023-11-18
**Revisions to the article**

We would like to thank the reviewers for providing constructive feedback. Based on that feedback, we have made the following changes:

1. In section 2, we make it clearer in the first paragraph that our paper does not propose a technique to train time series or causal machine learning models, but a methodology for RL evaluation and reliability on models that have already been trained. We also make it clear that we do not claim to be the first to use interventions in simulations for RL while including the citations provided [Reviewer mUDk]. We also differentiate our work from past RL reliability research at the end of the last paragraph of section 2 [Reviewer 8p4f].

2. In section 4, we made the article less verbose by moving subsection 4.1 to the appendix and leaving only the recommendations [Reviewer mUDk]. We also include a clear protocol that describes our methodology at the end of the section [Reviewer 8p4f].

3. In section 5, we include more thorough analyses of our experiments [Reviewer mUDk, 8p4f]. In particular, our Atari game analysis provides a nuanced look at the difference in robust performance between the Stable-Baselines3 A2C and PPO pretrained agents. We also assess multi-agent performance when agents of the original group are replaced by an outside group of agents of varying numbers and agent performance quality.

4. We included in the second paragraph of section 6 some of the weaknesses of our protocol [Reviewer okG3].

5. In sections 1 and 6, we emphasize the relevance of our methodology to recent developments in AI governance, like the EU's AI Act and the US Executive Order on safe use and development of AI. This inclusion is meant to further frame our paper as a RL reliability/robustness paper.

6. In the appendix, we added more information about the environments and experimental process. We also included the hyperparameters used for training the PowerGridworld agents.

Lastly, our code has to go through organizational approvals before it can be released. If our paper is accepted, we will seek approvals for an open source distribution version of our code and provide a link in the camera-ready version of the article.

---

### Meta-Review · Area_Chair_FyDv · 2023-12-07

**Metareview:**

This research tackles the reproducibility crisis in Reinforcement Learning (RL) evaluation, addressing issues like model overfitting and experimental design oversight. Despite existing reliability metrics, out-of-distribution observations are neglected. The proposed solution uses time series analysis tools to gauge RL model robustness amid distribution shifts. Applied in single-agent and multi-agent environments, these tools highlight the impact of introducing distribution shifts during test time, marking a crucial step towards a more rigorous RL evaluation framework.

This paper has some merits. Firstly, its comprehensive background on causal inference is presented in a way that aids newcomers without sacrificing readability. Secondly, its intuitive writing style enhances its potential adoption by the RL community. Lastly, the inclusion of a plot effectively highlights the limitations of relying solely on point estimates for model evaluation, contributing visually to the paper's overall impact.

On the other hand, the paper faces several criticisms: a lack of clarity in articulating the problem it aims to solve, a scattered methodology without clear application steps, inadequate result interpretation, and a related work section that lacks a clear distinction. Additionally, a notable weakness is the absence of any discussion in the main text on the potential drawbacks or limitations of the proposed method. Addressing these points would enhance the paper's clarity and contribution to the field.

Even if the authors' rebuttals have clarified some of the issues raised by the reviewers, the paper still needs significant improvement to be ready for publication. We encourage the authors to consider the reviewers' feedback while preparing a new version of their paper.

**Justification For Why Not Higher Score:**

The lack of clarity in articulating the problem and a vague problem formulation obscures the paper's purpose.
Inadequate result interpretation leaves uncertainties about agent robustness and algorithm performance. The related work section lacks a clear distinction from existing research.

**Justification For Why Not Lower Score:**

N/A

---

### Decision · Program_Chairs · 2024-01-16

Reject